# Integrated Surveillance of Disparities in Vaccination Coverage and Morbidity during the COVID-19 Pandemic: A Cohort Study in Southeast Sweden

**DOI:** 10.3390/vaccines12070763

**Published:** 2024-07-12

**Authors:** Armin Spreco, Örjan Dahlström, Dennis Nordvall, Cecilia Fagerström, Eva Blomqvist, Fredrik Gustafsson, Christer Andersson, Rune Sjödahl, Olle Eriksson, Jorma Hinkula, Thomas Schön, Toomas Timpka

**Affiliations:** 1Department of Health, Medicine, and Caring Sciences, Linköping University, 58183 Linköping, Sweden; armin.spreco@liu.se (A.S.); dennis.nordvall@rjl.se (D.N.); achristerandersson@gmail.com (C.A.); 2Regional Executive Office, Region Östergötland, 58225 Linköping, Sweden; rune.sjodahl@liu.se; 3Department of Behavioral Sciences and Learning, Linköping University, 58183 Linköping, Sweden; orjan.dahlstrom@liu.se; 4Qulturum Development Department, Region Jönköping County, 55305 Jönköping, Sweden; olle.eriksson@rjl.se; 5Department of Research, Region Kalmar County, 39185 Kalmar, Sweden; cecilia.fagerstrom@regionkalmar.se; 6Department of Computer and Information Science, Linköping University, 58183 Linköping, Sweden; eva.blomqvist@liu.se; 7Department of Electrical Engineering, Linköping University, 58183 Linköping, Sweden; fredrik.gustafsson@liu.se; 8Department of Biomedical and Clinical Sciences, Linköping University, 58183 Linköping, Sweden; jorma.hinkula@liu.se (J.H.); thomas.schon@liu.se (T.S.); 9Department of Infectious Diseases, County of Östergötland and Kalmar, Linköping University, 58183 Linköping, Sweden

**Keywords:** public health, medical informatics, health information systems, infectious diseases, COVID-19, epidemiology, health equity

## Abstract

We aimed to use the digital platform maintained by the local health service providers in Southeast Sweden for integrated monitoring of disparities in vaccination and morbidity during the COVID-19 pandemic. The monitoring was performed in the adult population of two counties (*n* = 657,926) between 1 February 2020 and 15 February 2022. The disparities monitored were relocated (internationally displaced), substance users, and suffering from a psychotic disorder. The outcomes monitored were COVID-19 vaccination, SARS-CoV-2 test results, and hospitalization with COVID-19. Relocated residents displayed an increased likelihood of remaining unvaccinated and a decreased likelihood of testing as well as increased risks of primary SARS-CoV-2 infection and hospitalization compared with the general population. Suffering from a major psychiatric disease was associated with an increased risk of remaining unvaccinated and an increased risk of hospitalization but a decreased risk of SARS-CoV-2 infection. From the digital monitoring, we concluded that the relocated minority received insufficient protection during the pandemic, suggesting the necessity for comprehensive promotion of overall social integration. Persons with major psychiatric diseases underused vaccination, while they benefitted from proactively provided testing, implying a need for active encouragement of vaccination. Further research is warranted on legal and ethical frameworks for digital monitoring in vaccination programs.

## 1. Introduction

Population health emergencies such as pandemics require rapid and coordinated, yet evidence-based, responses from health service providers. Because of the uneven impact of pandemics on national and local populations, response planning should include surveillance of all subpopulations to ensure that they are evenly reached by protective interventions. Health equity aims for social justice in health, while health disparities are indicators that can be employed to quantify progress towards achieving this goal [1]. Including health equity considerations in policymaking was deemed essential at the international, national, and local levels during the COVID-19 pandemic [2]. Modern digital health platforms offer health service providers the means to adapt local response strategies during global health emergencies [3]. However, the platforms do not always include functions for monitoring health disparities [4,5].

### 1.1. Motivation for the Initiation of Health Equity Surveillance during the COVID-19 Pandemic

At the start of the COVID-19 pandemic, it was not evident that severe morbidity was more widespread in vulnerable subpopulations. Some previous studies suggested that medical conditions have less prominent social gradients when less is known about prevention and treatment [6]. Yet other studies reported affluent groups to be among the first to be infected in Western countries [7,8]. Nonetheless, public health professionals soon cautioned that groups with limited control of working and living conditions were disadvantaged by higher risk of exposure to SARS-CoV-2, suggesting that factors such as occupation and income were candidate determinants of inferior pandemic outcomes [9,10,11]. In many countries, relocated (internationally displaced) residents who were forced to leave their country of origin because of armed conflict, religious persecution, or other types of oppression or deprivation face numerous disadvantages [12]. Limited control over living conditions, low trust in authorities, and language barriers contribute to lower uptake of health services in this group. For instance, residents relocated to Western Europe from Eastern Europe, Africa, the eastern Mediterranean, and Asia were considered to have an increased likelihood of being undervaccinated before the COVID-19 pandemic [13,14]. Additionally, early during the pandemic, it was reported that persons with major psychiatric diseases had an increased risk of COVID-19 hospitalization and/or death [15,16]. Failure to utilize health services increased their overall vulnerability, while physical inactivity, poor diet, and tobacco use, contributed to inferior outcomes from COVID-19 disease [17,18].

### 1.2. Swedish Public Health Strategies for the Pandemic Response

The first COVID-19 case in Sweden was reported on 8 February 2020. From the start of the pandemic to 15 February 2022, COVID-19 was defined as a reportable disease associated with a mandatory requirement for polymerase chain reaction (PCR) testing when symptoms were experienced. In accordance with its constitution, when only non-pharmaceutical interventions were available, Sweden pursued a “lower-scale” lockdown policy than most comparable countries. Although initially criticized [19], the protective effect of the less draconic policy showed to not differ significantly from that of more restrictive policies applied in comparable countries [20]. The detailed planning of the health service response was distributed to autonomous healthcare regions during the pandemic. All Swedish regions provided SARS-CoV-2 testing free of charge between 1 July 2020 and 15 February 2022. The COVID-19 vaccination program was initiated in late December 2020, and the first individuals were protected by 31 January 2021. The program was voluntary with the BNT162b2 vaccine used as the main vaccine [21,22].

No population faction was highlighted for more rigorous restrictions in the lockdown policy. Instead, some subgroups were excluded from the general restrictions as, for instance, elementary schools remained open. In the vaccination program, however, individuals 65 years of age or older and patients with chronic medical conditions associated with worse COVID-19 outcomes were defined as risk groups and prioritized for vaccination. Also, in Sweden, in the early stage of the pandemic, public health researchers cautioned that groups disadvantaged by limited control of working and living conditions were at higher risk of exposure to SARS-CoV-2 and lower rates of testing [23]. Relocated designation was identified as one of the most important disparities in this context [24]. Despite evidence of associated disadvantages, psychiatric disorders were not listed among the medical conditions qualifying for prioritization in the vaccination program.

### 1.3. The Digital Public Health Platform for Southeast Sweden

The Southeast Healthcare Region (SEHR) is the association of local government-financed healthcare providers delivering health services to the residents in Östergötland, Jönköping, and Kalmar Län counties in Southeast Sweden (total population 1,200,000). Before the COVID-19 pandemic, SEHR initiated the development of a digital health platform for service quality assessment and population health surveillance. As support for the management of population health emergencies, the platform included information resources for community surveillance, intervention design, and analyses of outcomes and impacts [25]. Among emergencies, the impact of hypothetical interventions, such as social distancing during pandemics, could be simulated using realistic population data and tools for capacity and needs analyses [26]. Before epidemic seasons or when the pandemic alert was issued by the WHO, local outbreak prediction algorithms could be calibrated using the most recent information about the infectious agent and local circumstances [27,28,29].

In the early stages of the COVID-19 pandemic, syndromic data on COVID-19 symptoms recorded in the platform from telenursing consultations were used to forecast local hospital capacity needs [29]. The digital health platform also included information for health equity surveillance. Based on Swedish legislation, the information available for routine monitoring of health disparities was restricted to data on sex, age, geographical area of residence, and country of birth.

### 1.4. Study Aims

Vaccination programs are seldom evaluated in a comprehensive public health context. To support the COVID-19 response in Southeast Sweden, the consortium of local health service providers employed a digital platform for integrated surveillance of health equity. We aimed to use the data routinely collected in the digital platform to monitor residents with relocated designation and persons with substance use or psychotic disorders for inequities in vaccination and morbidity. It was assumed that interactions among the vulnerabilities increased the risk for inferior COVID-19 outcomes and that monitoring reports adjusted for confounding factors differed from unadjusted reports.

## 2. Materials and Methods

Between 1 February 2020 and 15 February 2022, the digital public health platform at SEHR was used to collect pandemic equity data from the adult population in Östergötland and Jönköping counties (total population 850,000).

### 2.1. Study Data

The population monitored consisted of the 657,926 adults 18 years of age or older (49.4% women) residing in Östergötland and Jönköping counties on 1 February 2020.

### 2.2. Public Health Indicators (Outcomes)

We monitored vaccination coverage (received ≥ 2 doses of COVID-19 vaccine) from the start of the vaccination program. The primary health outcome monitored was admission to hospital with COVID-19. A COVID-19 diagnosis (ICD-10 U07.1) was confirmed by a PCR test for SARS-CoV-2. The primary health outcome data were divided into the period before vaccine protection (up to 31 January 2021) and the vaccination period (from 1 February 2021 until the end of this study). The secondary health outcome monitored was primary SARS-CoV-2 infection (PCR test positive for SARS-CoV-2 adjusted for testing coverage). Data on all PCR tests conducted at any hospital or test station in the study counties were retrieved with the last update on 15 February 2022.

### 2.3. Disparities (Exposures)

The risk of social disparity through international relocation was represented by the categories western-born and relocated. Residents born in Africa, Asia, South and Central America, and Eastern Europe were classified as relocated (Appendix A). The remaining unvaccinated was defined as having received fewer than the recommended two doses of the COVID-19 vaccine. The ICD-10 codes for a diagnosis of substance use or a psychiatric disorder with psychotic components recorded between 1 January 2019 and 15 February 2022 were grouped into two categories denoting (A) substance use (F10 Alcohol-related disorders, F11 Opioid-related disorders, F12 Cannabis-related disorders, F14 Cocaine-related disorders, F15 Other stimulant-related disorders, F16 Hallucinogen-related disorders, F18 Inhalant-related disorders, and F19 Other psychoactive substance-related disorders) and (B) psychotic disorders (F20 Schizophrenia, F23 Brief psychotic disorder, F25 Schizoaffective disorders, F28 Other psychotic disorder not due to a substance or known physiological condition, F29 Unspecified psychosis not due to a substance or known physiological condition, F30 Manic episode, F31 Bipolar disorder, and F32.3 Major depressive disorder, single episode, severe with psychotic features).

### 2.4. Data Analysis

Descriptive statistical methods were initially used to describe the study population regarding exposure to vulnerability through relocation, substance use, and suffering from a psychotic disorder. Pearson’s chi-squared tests were performed to compare the prevalence of substance use and psychotic disorders in the western-born and relocated residents, and the correlation coefficient (phi) was calculated to obtain the effect sizes; values < 0.1 denote weak effect; 0.1–0.3, low effect; 0.3–0.5, medium effect; and >0.5, large effect.

Crude rates of the monitored outcomes were then displayed for the total study population and separately for the vulnerable subpopulations. To determine bias-adjusted associations between the monitored exposures and outcomes, directed acyclic graphs (DAGs) were used to represent assumptions relating to the effects of exposures on the outcomes monitored (Appendix A). Informed by the DAGs, multiple binary logistic regression models were applied to estimate the effects of the main exposures relocated standing (1, yes; 0, no), substance use (1, yes; 0, no), and psychotic disorder (1, yes; 0, no), as well as their interactions on the outcomes remaining unvaccinated (1, yes; 0, no), primary SARS-CoV-2 infection (positive SARS-CoV-2 test at least once (1, yes; 0, no)), active SARS-CoV-2 testing (≥ 1 negative SARS-CoV-2 test recorded (1, yes; 0, no)), and hospitalization with COVID-19 before and during the vaccination period (1, yes; 0, no). The monitoring of testing positive for SARS-CoV-2 was adjusted for active testing, and vice versa. In the case of a significant interaction, the combinations with the main exposure were presented as simple main effects. All effects of each main exposure were adjusted for sex (1, male; 0, female), age, and the other main exposures. Furthermore, to monitor the associations between vaccination status and hospitalization during the vaccination period, exposure to remaining unvaccinated (1, yes; 0, no) was analyzed in three separate multiple models for the likelihood of hospitalization, one model for each vulnerability category. Nagelkerke R^2^ was obtained for all multiple models to estimate their accountability levels, and simple models were produced as a complement to support interpretation. The analyses were performed using the Statistical Package for the Social Sciences (SPSS) for Windows version 28.0.

A sensitivity analysis was applied to assess the robustness of the monitored exposure–outcome relationships to unmeasured confounding by calculating the E value (Appendix A) [30]. If the relationship between a main exposure and an outcome was statistically significant, the E value of the confidence limit closest to the null on an odds ratio (OR) scale (referred to as the E value limit) was used to evaluate its robustness. The E value limit represents the minimum strength of association that a hypothetical unmeasured confounder would need to have with both the exposure and the outcome to explain away the observed exposure–outcome relationship. Based on the available literature on exposures not accounted for in the adjusted monitoring (medical comorbidity, other socioeconomic factors, etc.) and COVID-19 outcomes [15,31,32], the E value limit was interpreted to be of small size in the range 1.20 to 1.39, of moderate size in the range 1.40 to 2.49, and of large size for limit values ≥2.50.

## 3. Results

### 3.1. Study Population

The mean age of the monitored population (*n* = 657,926) was 49.9 years (S.D. 19.6 years) and 51.0 years (S.D. 20.0 years) in women and 48.9 years (S.D. 19.2 years) in men (Table 1). About one adult person in six (16.8%, *n* = 110,451) had relocated to Sweden from a non-Western country (Appendix A). The mean age of the relocated subpopulation was 42.6 years (S.D. 15.6 years) and the western-born population was 51.4 years (S.D. 20.0 years). The most common countries of birth in the relocated minority (Syria (16.6%), Iraq (10.5%), Bosnia and Herzegovina (8.2%), and Somalia (5.6%)) had recently been involved in an armed conflict (Appendix A).

Regarding major psychiatric diseases, 2.2% (*n* = 14,262) of the adult population had a substance use diagnosis and 1.3% (*n* = 8387) had a diagnosis of a psychotic disorder. The western-born majority included a higher proportion of persons with a substance use diagnosis than the relocated minority (2.4%, *n* = 12,994 compared with 1.1%, *n* = 1268; *p* < 0.001; *phi* = 0.03) (Appendix A). Also, the proportion of persons with a psychotic disorder was larger in the western-born majority than in the relocated minority (1.3%, *n* = 7202 compared with 1.1%, *n* = 1185; *p* < 0.001; *phi* = 0.01) (Appendix A).

### 3.2. Vaccination Coverage

In the total adult population, 17.7% remained unvaccinated during the monitoring period, compared with 33.9% in persons with relocated status, 26.6% in persons diagnosed with substance use, and 25.4% in persons with a psychotic disorder (Table 2).

The adjusted monitoring of vaccination coverage showed that relocation (OR, 2.73; 95% confidence interval [CI], 2.69–2.77), substance use (OR, 1.88; 95% CI, 1.80–1.96), and having a psychotic disorder (OR, 1.68; 95% CI, 1.59–1.78)) were independently associated with an increased likelihood of remaining unvaccinated (Table 3). Interactions between relocated status and substance use (OR, 0.73; 95% CI, 0.65–0.83) and having a psychotic disorder (OR, 0.58; 95% CI, 0.51–0.67) were observed. The direction and magnitude of the simple effects of the interactions indicated that having any of the psychiatric disorders and relocated status was associated with a lower increase in the likelihood of remaining unvaccinated compared with having psychiatric disorders and being western-born. The odds ratios for the main effects in the simple models of not being vaccinated corresponded to the likelihoods observed in the multiple models (Appendix A).

### 3.3. Primary SARS-CoV-2 Infections

In the total adult population, 20.9% tested positive for SARS-CoV-2. In persons with relocated status, the prevalence of testing positive for SARS-CoV-2 was 25.4% compared with 15.5% in persons with substance use and 16.4% in persons with a psychotic disorder (Table 2).

The adjusted morbidity monitoring showed that relocation increased the likelihood of testing positive for SARS-CoV-2 (OR, 1.30; 95% CI, 1.28–1.32), while substance use (OR, 0.71; 95% CI, 0.67–0.74) and having a psychotic disorder (OR, 0.78; 95% CI, 0.73–0.83) decreased the likelihood (Table 3). No interaction between substance use and relocated status was observed, but there was an interaction between psychotic disorder and relocated status regarding testing positive (OR, 0.71; 95% CI, 0.60–0.84). The simple effects of the interaction indicated that having a psychotic disorder and relocated status was associated with a lower increase in the likelihood of testing positive for SARS-CoV-2 compared with having this condition and being western-born. The odds ratios for the main effects in the simple models of COVID-19 testing matched the likelihoods observed in the corresponding multiple models (Appendix A).

The prevalence of active SARS-CoV-2 testing was 51.7% in the total population, while it was 42.6% in persons with relocated status, 58.8% in persons diagnosed with substance use, and 59.0% in persons with a psychotic disorder (Table 2).

In the adjusted monitoring, relocation (OR, 0.51; 95% CI, 0.50–0.52) was associated with a lower likelihood of active testing for SARS-CoV-2, while substance use (OR, 1.38; 95% CI, 1.33–1.43) and having a psychotic disorder (OR, 1.25; 95% CI, 1.19–1.31) were associated with a higher likelihood (Table 4). Interactions between relocated status and both substance use (OR, 1.27; 95% CI, 1.13–1.44) and having a psychotic disorder (OR, 1.17; 95% CI, 1.03–1.33) were observed. The direction and magnitude of the simple effects of the interaction indicated that having any of the major psychiatric diseases and relocated status was associated with a higher increase in the likelihood of active SARS-CoV-2 testing compared with having these conditions and being western-born. The odds ratios for the main effects in the simple models of SARS-CoV-2 testing matched the likelihoods observed in the corresponding multiple models (Appendix A).

### 3.4. COVID-19 Hospitalizations

Less than 1 percent (0.9%) of the total population was hospitalized because of COVID-19 (Table 2). The proportion of COVID-19 hospitalizations in persons with relocated status was 1.6%, while 1.7% of persons with substance use and 2.1% of persons with a psychotic disorder were hospitalized during the monitoring period.

Regarding the adjusted monitoring during the pre-vaccination and vaccination periods, similar levels of notably increased likelihood were observed for relocated status (OR, 3.78; 95% CI 3.46–4.12 and OR, 3.03; 95% CI 2.79–3.28), substance use (OR, 2.13; 95% CI, 1.73–2.62 and OR, 2.13; 95% CI, 1.76–2.58), and psychotic disorders (OR, 3.16; 95% CI, 2.50–4.00 and OR, 2.25; 95% CI, 1.76–2.87) (Table 5). Relocated status showed a significant interaction during the pre-vaccination period with both substance use (OR, 0.42; 95% CI, 0.20–0.88) and having a psychotic disorder (OR, 0.34; 95% CI, 0.17–0.66). The direction and magnitude of the simple effects of this interaction indicated that having any of the psychiatric conditions and relocated status was associated with a lower increase in the likelihood of hospitalization compared with having these conditions and being western-born. In the vaccination period, no interactions were observed.

In the period when vaccination was available (Table 6), remaining unvaccinated was associated with a higher likelihood of hospitalization among the relocated (OR, 1.48; 95% CI, 1.30–1.70) and in persons with substance use (OR, 1.77; 95% CI, 1.23–2.55), while in persons with a psychotic disorder, the difference did not reach statistical significance (OR, 1.39; 95% CI, 0.89–2.18; *p* = 0.148). The odds ratios for the main effects in the simple models of hospitalization corresponded to the likelihoods observed in the multiple models (Appendix A).

### 3.5. Sensitivity Analyses

For the statistically significant relationships between the main exposures and the primary monitoring outcome hospitalization with COVID-19, an unmeasured confounder would have needed to have had a strong association with both the exposure and the outcome to explain away the lower confidence limit of the OR (E values ranging from 2.85 to 6.38). Regarding the secondary outcomes testing positive for SARS-CoV-2 and active SARS-CoV-2 testing, the E value limits for the exposure–outcome relationships were of moderate size (E values ranging from 1.41 to 2.12), while the E value limits for the outcome remaining unvaccinated were of moderate to large size (E values ranging from 1.83 to 2.66). The statistically significant relationships between the exposure to remaining unvaccinated and the outcome COVID-19 hospitalization showed E value limits of moderate size, ranging from 1.76 to 1.92.

## 4. Discussion

The digital monitoring of disparities in the adult population of two Swedish counties during the COVID-19 pandemic displayed generally inferior protection in the relocated minority. The influence of major psychiatric disease was more complex; a lower propensity for vaccination and higher propensity for SARS-CoV-2 testing was observed in parallel with decreased risk of SARS-CoV-2 infection and increased risk of hospitalization. Interactions among the disparities contributed to a lower-than-expected exacerbation of COVID-19 outcomes. Sensitivity analyses suggested that the total unmeasured confounding was not a plausible reason for explaining away these findings.

The generally inferior pandemic outcomes observed in the relocated minority can be interpreted to reflect structural disparity, i.e., a disadvantaged hierarchical position of the relocated subpopulation in the local community [33]. Corroborating data reported from an elderly Swedish population [34], our monitoring showed that the likelihood of remaining unvaccinated was higher and the likelihood of active SARS-CoV-2 testing was lower in the relocated minority compared with the western-born majority. These undesirable associations between relocated status and pandemic protection are likely to reflect general challenges experienced by populations relocated to countries with Western healthcare and welfare state systems. For example, relocated residents may find written formats inappropriate and prefer oral messaging in their own language [35,36]. In the absence of accessible information sources, they are more susceptible to mis- and disinformation because they must rely on friends, family, and unregulated resources, such as social media, for advice on health protection [37,38]. However, the integrated monitoring also showed that individuals with relocated status and a major psychiatric disease, who thus were recognized by the local healthcare providers, had a lower-than-expected likelihood of remaining unvaccinated and a higher-than-expected likelihood of active testing. Such mitigating interactions have also been reported in other settings [39,40]. The monitoring findings thus suggest that healthcare contacts provided an interface to the mainstream local community for the relocated population faction.

Explanations of the observed decreased likelihood of primary SARS-CoV-2 infection in persons with major psychiatric diseases include a lower exposure to the pandemic virus resulting from social isolation and hygienic vigilance in healthcare settings [41,42,43]. However, despite the decreased risk for primary infection, the risk of COVID-19 hospitalization was increased in these individuals. Apathy and other aspects of the psychiatric disease may have delayed seeking health care when experiencing symptoms of COVID-19, leading to delayed treatment. Failure to seek healthcare has been reported to increase overall vulnerability in psychiatric patients during the pandemic [17,44,45]. Although psychiatric diseases are reported to increase the risk of a poor clinical outcome in several physical conditions [46], medical co-morbidity is a less likely explanation of the findings the monitoring data supports. Having a major psychiatric disease was associated with decreased vaccination coverage but increased SARS-CoV-2 testing, indicating that more likely explanations of the inferior pandemic outcomes are apathy and social isolation [47]. While vaccination was voluntary and necessitated a basic level of self-efficacy in physical health management, vigilant SARS-CoV-2 testing was included in the local healthcare routines and required no personal initiative. Consequently, low self-efficacy in physical health management may have been a key reason for the observed inferior pandemic outcomes in persons with major psychiatric diseases. Active encouragement of vaccination by general practitioners and family caretakers has been shown to increase vaccination rates in persons with psychotic disorders at medical risk [48]. The interpretation that self-efficacy played a central role in preventing COVID-19 hospitalizations in persons with major psychiatric conditions led to the inference that the collaborative care model used in Southeast Sweden needed to be reconsidered with respect to physical health management [49]. Regarding COVID-19, self-efficacy is particularly important considering that oral antiviral treatment needs to be initiated within 5 days after the first symptoms in order to prevent severe disease [50].

This study is unique in that a digital platform was used to monitor pandemic protection in local vulnerable minorities identifiable by routinely accessible data. The size of the relocated population faction and the prevalence of major psychiatric diseases in the monitored population are representative of Sweden overall. The monitoring was based on registry-based data of high quality and completeness. The population, vaccination, testing, and diagnostic data were comprehensive and prospectively collected because COVID-19 reporting was mandatory during the monitoring period, resulting in minimal selection bias. We report both crude monitoring data and monitoring estimates adjusted for variations in SARS-CoV-2 testing propensity in different population strata. Uncontrolled confounding was accounted for in the sensitivity analysis. Nonetheless, bias may remain in the results. For instance, we did not include pre-hospital mortality as a competing risk in the analysis of morbidity. Because pre-hospital COVID-19 mortality is a relatively rare event, this would not change the monitoring results but could lead to a slight overestimation of hospitalizations. More types of social disparities, such as unemployment and poor housing conditions, could also have been adjusted for in the analyses, but such data were not available in the digital platform for legal reasons. However, a recent study of the impact of social disparities on COVID-19 morbidity in Sweden showed that relocation status was the most important disparity in the pandemic context [24]. Moreover, the term “relocated” could have been defined differently. A limitation of this study is the dichotomizing of the country of birth for scientific reasons. Residents born in some non-Western countries were designated as “relocated” although their countries are very close to the West, e.g., the Baltic republics and the Czech Republic. When interpreting the results, it should therefore be considered that the individuals designated as relocated originated from countries ranging from Syria and Somalia to new EU members. Nonetheless, we find the division into Western and non-Western countries of birth is justified based on recent data on pandemic response collected from Western Europe [51] and taking into consideration factors such as the collective experiences of corruption and trust in authorities [52]. Finally, a lateral explanation of the increased likelihood of COVID-19 hospitalization in persons with major psychiatric diseases is that they required inpatient treatment on a regular basis for their psychiatric illness, i.e., that the increased rates include patients testing positive for SARS-CoV-2 when primarily hospitalized for a psychiatric disease [53].

## 5. Conclusions

Vaccination programs are too seldom evaluated in a comprehensive public health context. Digital public health monitoring in Southeast Sweden during the COVID-19 pandemic revealed that the local relocated minority received insufficient protection, suggesting the necessity for promoting overall social integration [54,55]. While persons with major psychiatric diseases underused vaccination, they benefitted from proactively provided testing, implying that vaccination should also be actively encouraged in this group. These results suggest that a digital platform incorporating health equity surveillance is an essential resource in local pandemic preparedness [4,5]. Further research is warranted on legal and ethical frameworks for routine digital monitoring of health disparities.

## Figures and Tables

**Table 1 vaccines-12-00763-t001:** Distribution in numbers (percent of all adults) of age and sex categories in the monitored population in Östergötland and Jönköping counties, Sweden, (*n* = 657,926) and corresponding subpopulations disaggregated by global region of birth.

Age Category	Women, *n* (%)	Men, *n* (%)	Total, *n* (%)
18–39 years	110,242 (16.8)	123,313 (18.7)	233,555 (35.5)
Western-born	86,223 (13.1)	94,694 (14.4)	180,917 (27.5)
Relocated	24,019 (3.7)	28,619 (4.3)	52,638 (8.0)
40–64 years	122,013 (18.5)	127,311 (19.4)	249,324 (37.9)
Western-born	98,135 (14.9)	104,314 (15.9)	202,449 (30.8)
Relocated	23,878 (3.6)	22,997 (3.5)	46,875 (7.1)
65–79 years	64,567 (9.8)	62,328 (9.5)	126,895 (19.3)
Western-born	60,010 (9.1)	58,024 (8.8)	118,034 (17.9)
Relocated	4557 (0.7)	4304 (0.7)	8861 (1.3)
80+ years	28,368 (4.3)	19,784 (3.0)	48,152 (7.3)
Western-born	27,149 (4.1)	18,926 (2.9)	46,075 (7.0)
Relocated	1219 (0.2)	858 (0.1)	2077 (0.3)
All adults	325,190 (49.4)	332,736 (50.6)	657,926 (100.0)
Western-born	271,517 (41.3)	275,958 (41.9)	547,475 (83.2)
Relocated	53,673 (8.2)	56,778 (8.6)	110,451 (16.8)

**Table 2 vaccines-12-00763-t002:** Distribution of crude monitoring outcomes among adult residents in Östergötland and Jönköping counties, Sweden, between 1 February 2020 and 15 February 2022 considering relocation status and major psychiatric conditions.

	*n*	Remaining Unvaccinated, *n* (%)	Positive SARS-CoV-2 Test (≥1 Positive Test), *n* (%)	Active Testing (≥1 Negative Test), *n* (%)	Hospitalization (≥1), *n* (%)
Total population	657,926	116,637 (17.7)	137,268 (20.9)	339,841 (51.7)	6027 (0.9)
Relocation category					
Western-born	547,475	79,167 (14.5)	109,196 (19.9)	292,745 (53.5)	4235 (0.8)
Relocated	110,451	37,470 (33.9)	28,072 (25.4)	47,096 (42.6)	1792 (1.6)
Persons with major psychiatric diseases					
Substance use					
All	14,262	3759 (26.4)	2206 (15.5)	8392 (58.8)	242 (1.7)
Western-born	12,994	3224 (24.8)	1973 (15.2)	7704 (59.3)	216 (1.7)
Relocated	1268	535 (42.2)	233 (18.4)	688 (54.3)	26 (2.1)
Psychotic disorder					
All	8387	2128 (25.4)	1379 (16.4)	4949 (59.0)	172 (2.1)
Western-born	7202	1730 (24.0)	1196 (16.6)	4346 (60.3)	143 (2.0)
Relocated	1185	398 (33.6)	183 (15.4)	603 (50.9)	29 (2.4)

**Table 3 vaccines-12-00763-t003:** Adjusted estimates of associations among remaining unvaccinated and relocation status, substance use, having a psychotic disorder, and their interactions among adult residents in Östergötland and Jönköping counties, Sweden, 1 February 2020 to 15 February 2022.

	Remaining Unvaccinated OR (95% CI) (*p* Value) [E Value Limit]
Main exposures	
Relocated (1, yes; 0, no)	2.73 (2.69–2.77) (<0.001) [2.66]
Substance use (1, yes; 0, no)	1.88 (1.80–1.96) (<0.001) [2.02]
Psychotic disorder (1, yes; 0, no)	1.68 (1.59–1.78) (<0.001) [1.83]
Interactions	
Substance use by relocation status	0.73 (0.65–0.83) (<0.001)
Relocated–substance use (1, yes; 0, no)	1.38 (1.23–1.54) (<0.001)
Western-born–substance use (1, yes; 0, no)	1.88 (1.80–1.96) (<0.001)
Substance use–relocated (1, yes; 0, no)	2.00 (1.77–2.26) (<0.001)
Psychotic disorder by relocation status	0.58 (0.51–0.67) (<0.001)
Relocated–psychotic disorder (1, yes; 0, no)	0.98 (0.87–1.12) (0.773)
Western-born–psychotic disorder (1, yes; 0, no)	1.68 (1.59–1.78) (<0.001)
Psychotic disorder–relocated (1, yes; 0, no)	1.59 (1.39–1.82) (<0.001)
Moderators	
Sex (1, male; 0, female)	1.02 (1.01–1.03) (0.002)
Age (continuous)	0.98 (0.98–0.98) (<0.001)
Nagelkerke R^2^	0.08

Odds ratios (multiple logistic regression models) represent associations between substance use and psychotic disorder and the likelihood of remaining unvaccinated. The lower limit of the E values for the main exposures is presented for significant associations. For significant interactions, the total effects (interactions combined and simple effects) are shown. *p* values > 0.05 are not significant.

**Table 4 vaccines-12-00763-t004:** Adjusted estimates of associations among active testing (≥1 Negative Test) and positive SARS-CoV-2 tests (≥1) and relocation status, substance use, having a psychotic disorder, and their interactions among adult residents in Östergötland and Jönköping counties, Sweden, from 1 February 2020 to 15 February 2022.

	Positive SARS-CoV-2 Test OR (95% CI) (*p* Value) [E Value Limit]	Active SARS-CoV-2 Testing OR (95% CI) (*p* Value) [E Value Limit]
Main exposures		
Relocated (1, yes; 0, no)	1.30 (1.28–1.32) (<0.001) [1.52]	0.51 (0.50–0.52) (<0.001) [2.12]
Substance use (1, yes; 0, no)	0.71 (0.67–0.74) (<0.001) [1.60]	1.38 (1.33–1.43) (<0.001) [1.57]
Psychotic disorder (1, yes; 0, no)	0.78 (0.73–0.83) (<0.001) [1.43]	1.25 (1.19–1.31) (<0.001) [1.41]
Interactions		
Substance use by relocation status	0.92 (0.79–1.08) (0.309)	1.27 (1.13–1.44) (<0.001)
Relocated–substance use (1, yes; 0, no)		1.76 (1.56–1.97) (<0.001)
Western-born–substance use (1, yes; 0, no)		1.38 (1.33–1.43) (<0.001)
Substance use–relocated (1, yes; 0, no)		0.65 (0.57–0.73) (<0.001)
Psychotic disorder by relocation status	0.71 (0.60–0.84) (<0.001)	1.17 (1.03–1.33) (0.015)
Relocated–psychotic disorder (1, yes; 0, no)	0.55 (0.47–0.65) (<0.001)	1.47 (1.30–1.65) (<0.001)
Western-born–psychotic disorder (1, yes; 0, no)	0.78 (0.73–0.83) (<0.001)	1.25 (1.19–1.31) (<0.001)
Psychotic disorder–relocated (1, yes; 0, no)	0.92 (0.77–1.09) (0.348)	0.60 (0.53–0.68) (<0.001)
Moderators		
Sex (1, male; 0, female)	0.82 (0.81–0.83) (<0.001)	0.64 (0.63–0.64) (<0.001)
Age (continuous)	0.98 (0.98–0.98) (<0.001)	0.98 (0.98–0.98) (<0.001)
Testing indicators		
Active testing (≥1 negative test) (1, yes; 0, no)	1.99 (1.96–2.01) (<0.001)	-
Tested positive (≥1) (1, yes; 0, no)	-	1.97 (1.95–2.00) (<0.001)
Nagelkerke R^2^	0.08	0.10

Odds ratios (multiple logistic regression models) represent associations between substance use and psychotic disorder and the likelihood of COVID-19 monitoring (≥1 negative RT-qPCR test) and having tested positive for SARS-CoV-2 (≥1 positive RT-qPCR test). The likelihood of COVID-19 monitoring was corrected for having tested positive for SARS-CoV-2, and vice versa. The lower limit of the E values for the main exposures is presented for significant associations. For significant interactions, the total effects (interactions combined and simple effects) are shown. *p* values > 0.05 are not significant.

**Table 5 vaccines-12-00763-t005:** Adjusted estimates of associations among a hospitalization with COVID-19 and relocation status, substance use, having a psychotic disorder, and their interactions among adult residents in Östergötland and Jönköping counties, Sweden, before vaccination was available (February 2020 to January 2021) and in the vaccination period (February 2021 to February 2022).

	Hospitalization with COVID-19 (*n* = 657,926)	
	before Vaccination OR (95% CI) (*p* Value) [E Value Limit]	in Vaccination Period OR (95% CI) (*p* Value) [E Value Limit]
Main exposures		
Relocated (1, yes; 0, no)	3.78 (3.46–4.12) (<0.001) [6.38]	3.03 (2.79–3.28) (<0.001) [5.02]
Substance use (1, yes; 0, no)	2.13 (1.73–2.62) (<0.001) [2.85]	2.13 (1.76–2.58) (<0.001) [2.92]
Psychotic disorder (1, yes; 0, no)	3.16 (2.50–4.00) (<0.001) [4.44]	2.25 (1.76–2.87) (<0.001) [2.92]
Interactions		
Substance use by relocation status	0.42 (0.20–0.88) (0.022)	0.73 (0.44–1.23) (0.239)
Relocated–substance use (1, yes; 0, no)	0.90 (0.45–1.83) (0.779)	
Western-born–substance use (1, yes; 0, no)	2.13 (1.73–2.62) (<0.001)	
Substance use:	1.60 (0.77–3.32) (0.205)	
Psychotic disorder by relocation status	0.34 (0.17–0.66) (0.001)	0.78 (0.47–1.31) (0.344)
Relocated–psychotic disorder (1, yes; 0, no)	1.06 (0.57–2.00) (0.852)	
Western-born–psychotic disorder (1, yes; 0, no)	3.16 (2.50–4.00) (<0.001)	
Psychotic disorder–relocated (1, yes; 0, no)	1.27 (0.65–2.48) (0.486)	
Moderators		
Sex (1, male; 0, female)	1.25 (1.16–1.35) (<0.001)	1.27 (1.18–1.36) (<0.001)
Age (continuous)	1.05 (1.04–1.05) (<0.001)	1.03 (1.03–1.03) (<0.001)
Nagelkerke R^2^	0.07	0.04

Odds ratios (multiple logistic regression models) represent associations between hospitalization with COVID-19 and substance use and psychotic disorder during two periods: pre-vaccination (up to 31 January 2021) and during vaccination (from 1 February to 2021 and up to 15 February 2022). The lower limit of the E values for the main exposures is presented for significant associations. The total effects (interactions combined and simple effects) are shown for significant interactions. *p* values > 0.05 are not significant.

**Table 6 vaccines-12-00763-t006:** Adjusted estimates of associations in the vaccination period between hospitalization with COVID-19 and remaining unvaccinated in the subpopulations with relocated status, substance use, and psychotic disorders.

	Hospitalization with COVID-19 in Vaccination Period among Residents with		
	Relocated (*n* = 110,451)	Substance Use (*n* = 14,262)	Psychotic Disorder (*n* = 8387)
	OR (95% CI) (*p* Value) [E-Value Limit]	OR (95% CI) (*p* Value) [E Value Limit]	OR (95% CI) (*p* Value) [E Value Limit]
Main exposure			
Remaining unvaccinated (1, yes; 0, no)	1.48 (1.30–1.70) (<0.001) [1.92]	1.77 (1.23–2.55) (0.002) [1.76]	1.39 (0.89–2.18) (0.148) [NA]
Main exposures adjusted and moderators			
Relocated (1, yes; 0, no)	-	1.82 (1.08–3.06) (0.024)	1.85 (1.11–3.09) (0.018)
Substance use (1, yes; 0, no)	1.58 (0.98–2.55) (0.061)	-	1.68 (1.01–2.80) (0.046)
Psychotic disorder (1, yes; 0, no)	1.74 (1.10–2.73) (0.018)	1.98 (1.24–3.18) (0.005)	-
Sex (1, male; 0, female)	1.30 (1.14–1.48) (<0.001)	1.11 (0.76–1.61) (0.585)	1.09 (0.71–1.67) (0.690)
Age (continuous)	1.04 (1.04–1.05) (<0.001)	1.03 (1.02–1.04) (<0.001)	1.01 (1.00–1.03) (0.049)
Nagelkerke R^2^	0.05	0.03	0.02

Odds ratios (multiple logistic regression models) represent associations between hospitalization with COVID-19 and vaccination status among substance use patients and patients with psychotic disorders when vaccination was administrated (after 31 January 2021). The lower limit of the E values for the main exposures is presented for significant associations. *p* values > 0.05 are not significant.

## Data Availability

The datasets generated and/or analyzed during the current study are not publicly available because of Swedish legislation regarding the management of research data but are available from the corresponding author upon reasonable request.

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
