# Peer review of "Integrated Surveillance of Disparities in Vaccination Coverage and Morbidity during the COVID-19 Pandemic: A Cohort Study in Southeast Sweden"

_vaccines, 2024, doi:10.3390/vaccines12070763_

Round 1

Reviewer 1 Report

Comments and Suggestions for Authors

Dear Authors,

Your article is very interesting and reveals important public health information.

However, I have several comments:

Correlation coefficient (φ) could be expressed as phi.

Within the results section the description of the study population should not be limited only to the variables of interest. Please add a brief description of their demographics as well including the age.

The results are very well described. It is very useful that you added the coding of the variables and makes the results clear and easy to be understood.

Adding the supplementary materials also allows the readers to understand the entire context of the article.

Strength of your study is that it covers large sample and is based on structured and reliable database.

One of the major limitations of your study is dichotomizing the country of origin. East in fact is stratified to different subtypes and some of them are very close to the west. For instance these are the Baltic republics, Check republic etc. These persons are much more likely to integrate like the west-born foreigners despite the different cultures like African and Asian-born citizens. Moreover, you cannot compare any relocated inhabitants originated from the new EU members to those from Afghanistan or Somalia; they are very different. It is worthy to be mentioned in the article that you dichotomized so different relocated inhabitants for scientific reasons.

Comments on the Quality of English Language

I would suggest birth country to be replaced with country of birth

Author Response

Comments and Suggestions for Authors

Dear Authors,

Your article is very interesting and reveals important public health information.

However, I have several comments:

Correlation coefficient (φ) could be expressed as phi.

Authors’ response:  Thank you! The change has been made.

Within the results section the description of the study population should not be limited only to the variables of interest. Please add a brief description of their demographics as well including the age.

The results are very well described. It is very useful that you added the coding of the variables and makes the results clear and easy to be understood.

Authors’ response:  Thank you. We have included a new table describing the demographics of the study population (Table 1 in the revised manuscript).

Adding the supplementary materials also allows the readers to understand the entire context of the article.

Authors’ response:  Thank you!

Strength of your study is that it covers large sample and is based on structured and reliable database.

One of the major limitations of your study is dichotomizing the country of origin. East in fact is stratified to different subtypes and some of them are very close to the west. For instance these are the Baltic republics, Check republic etc. These persons are much more likely to integrate like the west-born foreigners despite the different cultures like African and Asian-born citizens. Moreover, you cannot compare any relocated inhabitants originated from the new EU members to those from Afghanistan or Somalia; they are very different. It is worthy to be mentioned in the article that you dichotomized so different relocated inhabitants for scientific reasons.

Authors’ response:  Thank you. This is an important point. We have extended the discussion of the study limitations (lines 87-95) to highlight that we dichotomized the inhabitants into relocated (non-western born) and non-relocated (western born) for scientific reasons.

Comments on the Quality of English Language

I would suggest birth country to be replaced with country of birth

Authors’ response:  Thank you. The suggested change has been made.

Reviewer 2 Report

Comments and Suggestions for Authors Comments and suggestions Title section: 1. Title: public health effectiveness in the title seems too broad for the specific objective of this study Summary section: 2. The summary does not explicitly indicate the objective of the research. 3. Quantitative results have not been posted to give an idea of ​​the magnitude of the results. Introduction section: 4. The objectives should be clearer and eliminate redundancy in the paragraph where it is explicitly stated. Materials and methods section: 5. In the study data, the monitored population does not coincide with what is indicated in the summary or specify that it only refers to those over 18 years of age. 6. Indicate why these study populations were chosen: psychotics, diagnosis of substance use. 7. Explain the absence of a control group. Results Section: 8. This section would benefit from a table indicating the quantity that represents each subpopulation studied. Discussion Section: 9. Restructure paragraphs 1 and two based on the objective of the study. Conclusions Section: 10. Restructure conclusions based on the objectives of the study                               Comments on the Quality of English Language

Minor editing of English language required

Author Response

Comments and Suggestions for Authors

Title section: 1. Title: public health effectiveness in the title seems too broad for the specific objective of this study

Authors’ response: We do agree. Mention of public health effectiveness has been excluded from the title.

Summary section:

  1. The summary does not explicitly indicate the objective of the research.

Authors’ response:  Thank you. The objective of the research has been stated in the Summary.

  1. Quantitative results have not been posted to give an idea of the magnitude of the results.

Authors’ response:  Thank you for the comment. We would have been happy to include quantitative results in the summary but the word limit (200 words) disallowed this.

Introduction section:

  1. The objectives should be clearer and eliminate redundancy in the paragraph where it is explicitly stated.

Authors’ response:  The formulation of the objectives has been revised as suggested and redundancy eliminated.

Materials and methods section:

  1. In the study data, the monitored population does not coincide with what is indicated in the summary or specify that it only refers to those over 18 years of age.

Authors’ response:  Thank you for the comment. The description of study population has been revised as suggested.

  1. Indicate why these study populations were chosen: psychotics, diagnosis of substance use.

Authors’ response:  The choice of disadvantaged subpopulation for the study was informed by 1) estimation of risk for suffering disadvantages during the pandemic, and 2) data availability. The choice of persons with major psychiatric diseases (diagnosis of psychosis or substance use) was based on worrying reports early during the pandemic (see references 15 and 16). This has been made clearer in the revised manuscript.

  1. Explain the absence of a control group.

Authors’ response:  This was an observational total population study. The non-exposed population factions were in the analyses used as controls to those exposed.

Results Section:

  1. This section would benefit from a table indicating the quantity that represents each subpopulation studied.

Authors’ response:  Thank you! We totally agree and have included a table and more sociodemographic information in the Results section.

Discussion Section:

  1. Restructure paragraphs 1 and two based on the objective of the study.

Authors’ response:  The first two paragraphs in the Results section have been restructured.

Conclusions Section:

  1. Restructure conclusions based on the objectives of the study

Authors’ response:  The Conclusion has been restructured to better answer to the objectives of the study.    

Reviewer 3 Report

Comments and Suggestions for Authors

Introduction

1) Sentences in lines 47-50, 59-61, 62-63, are not clear what the authors intent to said. Please, contact a profesional translation service. 

2) Sentence in line 67-69, seems not to be related with the COVID-19 pandemic. In fact, he second paragraph of the introduction doesn't make much sense; it's a collection of disjointed ideas with difficult wording to understand.

3) Line 81-82, the molecular technique to detect SARS-CoV-2 virus was not reverse-transcription polymerase chain reaction, it was real time reverse-transcription polymerase chain reaction (is not the same). Also this technique is not PCR is RT-qPCR (those are very different techniques).

4) The redaction of all introduction section is very difficult to understand. 

5) Introduction is too long (with a lot of not related ideas and unorganized), please summarize what they want to express.

6) Line 135, what is SEHR?

7) Redaction of results is very chaotic. There is not a table that reflect the sociodemographic characteristic of the population studied. 

8) The study seems to be a big mixture of several projects. 

Comments on the Quality of English Language

They need a professional translate system. 

Author Response

Comments and Suggestions for Authors

Introduction

1) Sentences in lines 47-50, 59-61, 62-63, are not clear what the authors intent to said. Please, contact a profesional translation service.

Authors’ response:  Thank you. The sentences have been revised for clarity by an English language consultant.

2) Sentence in line 67-69, seems not to be related with the COVID-19 pandemic. In fact, he second paragraph of the introduction doesn't make much sense; it's a collection of disjointed ideas with difficult wording to understand.

Authors’ response:  The second paragraph has been revised for clarity by an English language consultant.

3) Line 81-82, the molecular technique to detect SARS-CoV-2 virus was not reverse-transcription polymerase chain reaction, it was real time reverse-transcription polymerase chain reaction (is not the same). Also this technique is not PCR is RT-qPCR (those are very different techniques).

Authors’ response:  Thank you. The sentence has been revised.

4) The redaction of all introduction section is very difficult to understand.

Authors’ response:  The Introduction section has been revised for clarity using guidance from an English language consultant.

5) Introduction is too long (with a lot of not related ideas and unorganized), please summarize what they want to express.

Authors’ response:  The objectives of the study have been more distinctly stated and the entire Introduction section has been thoroughly revised for clarity.

6) Line 135, what is SEHR?

Authors’ response:  The explanation was provided in lines 104-6: ‘The Southeast Healthcare Region (SEHR) is the association of the local-government financed healthcare providers delivering health services to the residents in Östergötland, Jönköping, and Kalmar Län counties in Southeast Sweden (total population 1.200,000).’

7) Redaction of results is very chaotic. There is not a table that reflect the sociodemographic characteristic of the population studied.

Authors’ response:  Thank you. We agree and have included an additional table and more sociodemographic information in the first two paragraphs of the Results section.

8) The study seems to be a big mixture of several projects.

Authors’ response:  Thank you. This was one integrated project. The study objectives and research questions were directly derived from the tasks and challenges that had to be managed in local public health work during the pandemic (in which most of the authors were involved). The study thus reports knowledge of relevance for hands-on vaccination practice (to be compared with theoretical evidence, such as vaccine efficacy).

Round 2

Reviewer 2 Report

Comments and Suggestions for Authors

I have no further comments or suggestions about the manuscript. The authors have responded to my previous comments to the best of their ability.

Comments on the Quality of English Language

Minor editing of English language required

Author Response

Reviewer 2

I have no further comments or suggestions about the manuscript. The authors have responded to my previous comments to the best of their ability.

Authors’ reply:  Many thanks for your constructive comments that have helped us to improve the quality of the manuscript.

Reviewer 3 Report

Comments and Suggestions for Authors

1) De authors did not modified PCR by RT-qPCR (lines 270-278).

2) There are still some minors mistakes in grammar. 

3) Tables 3, 4, 5 and 6, add beta value, r square, intersection, and p value of the regression.

4)

Comments on the Quality of English Language

minor corrections

Author Response

Reviewer 3

1) De authors did not modified PCR by RT-qPCR (lines 270-278).

Authors’ reply:  Thank you. The suggested addition has been made,

2) There are still some minors mistakes in grammar.

Authors’ reply: Thank you. We have again scanned and corrected the manuscript for mistakes in presentation, formatting, and grammar.

3) Tables 3, 4, 5 and 6, add beta value, r square, intersection, and p value of the regression.

 Authors’ reply: Many thanks for the constructive comment. We have added information on Nagelkerke R2 to tables 3-6.  The p-values were already presented. The beta values ​​do not add information as they ​​can easily be calculated using odds ratios (Odds Ratio = EXP(beta), or conversely Beta=LN(Odds Ratio)).